# GAG Protein of *Arabidopsis thaliana* LTR Retrotransposon Forms Retrosome-like Cytoplasmic Granules and Activates Stress Response Genes

**DOI:** 10.3390/plants14131894

**Published:** 2025-06-20

**Authors:** Alexander Polkhovskiy, Roman Komakhin, Ilya Kirov

**Affiliations:** 1Moscow Center for Advanced Studies, Kulakova Str. 20, 123592 Moscow, Russia; polkhovsky.a.w@gmail.com (A.P.); komakhin@gmail.com (R.K.); 2All-Russia Research Institute of Agricultural Biotechnology, Timiryazevskaya Str. 42, 127550 Moscow, Russia

**Keywords:** LTR retrotransposon, GAG, RNA-Seq, Cellular aggregates, RNAseq

## Abstract

LTR retrotransposons are widespread genomic elements that significantly impact genome structure and function. In *Arabidopsis thaliana*, the EVD LTR retrotransposon encodes a GAG protein essential for retrotransposon particle assembly. Here, we present a comprehensive analysis of the structural features, intracellular localization, and transcriptomic effects of the EVD GAG (evdGAG) protein. Using AlphaFold3, we identified canonical capsid (CA-NTD and CA-CTD) and nucleocapsid (NC) domains, with predicted disordered regions likely facilitating oligomerization. Transient expression of GFP-tagged evdGAG in protoplasts of *A. thaliana* and distant plant species (*Nicotiana benthamiana* and *Helianthus annuus*) revealed the formation of multiple large cytoplasmic aggregates resembling retrosomes, often localized near the nucleus. Stable overexpression of evdGAG in wild-type and *ddm1* mutant backgrounds induced significant transcriptomic changes, including up-regulation of stress response and defense-related genes and downregulation of photosynthesis and chloroplast-associated pathways. Importantly, genes linked to stress granule formation were also up-regulated, suggesting a role for evdGAG in modulating cellular stress responses. Our findings provide novel insights into the cellular and molecular properties of plant retrotransposon GAG proteins and their influence on host gene expression.

## 1. Introduction

Transposable elements (TEs) are key drivers of genetic diversity across the tree of life [1,2]. Their activity can profoundly reshape genome architecture, as exemplified by bread wheat (*Triticum aestivum*), where TEs constitute up to 85% of the genome [3]. TEs are broadly classified into two groups: Class II DNA transposons, which propagate via a “cut-and-paste” mechanism during DNA replication, and Class I retrotransposons (RTs), which employ a “copy-and-paste” strategy involving RNA intermediates [4]. RTs are further subdivided into Long Terminal Repeat (LTR) and non-LTR retrotransposons, with LTR retrotransposons being the most prevalent TEs in plant genomes [5]. LTR retrotransposons encode essential structural and enzymatic proteins (GAG, protease (PR), integrase (INT), and reverse transcriptase-RNase H (RT-RNase) to complete their replication cycle [5]. The reverse transcriptase reaction occurs within virus-like particles (VLPs). The VLPs are formed from numerous GAG proteins and serve as a capsule, protecting RT RNA [6,7]. GAG molar excess is a critical regulatory challenge in the retrotransposon life cycle, as structural GAG proteins must outnumber enzymatic POL components to ensure proper VLP assembly [8]. LTR retrotransposons have evolved diverse mechanisms to achieve this balance. In *Arabidopsis thaliana*, the EVD retrotransposon produces two RNA isoforms via alternative splicing: a truncated “short GAG” (shGAG) transcript and a full-length GAG-POL transcript. Splicing of the PR intron induces a frameshift that terminates GAG translation, favoring shGAG production. The shGAG isoform is associated with polysomes enabling a high rate of GAG protein translation [9].

VLP biogenesis depends on GAG multimerization through protein–protein interactions and RNA binding, a conserved feature shared by retroviruses and retrotransposons [10,11]. The GAG capsid (CA) domain mediates oligomerization, with critical residues identified through HIV-1 mutagenesis studies [12]. Remarkably, CA and nucleocapsid domains show deep evolutionary conservation across *Retroviridae*, *Metaviridae*, *Pseudoviridae*, *Caulimoviridae*, and *Hepadnaviridae* and key amino acids mentioned above are located within highly conserved regions across numerous viral species [13]. GAG protein is a main component of retrosomes, which are specialized cytoplasmic foci primarily described in yeast (*Saccharomyces cerevisiae*) [14,15,16]. Retrosomes serve as sites for VLP assembly, where retrotransposon GAG proteins multimerize around genomic RNA [17]. This process ensures the proper packaging of genomic RNA (gRNA) and enzymatic POL components. In Ty1 and Ty3 retrotransposons, retrosomes coordinate the transition of genomic RNA from a translation template to a packaged genome, mediated by interactions with host factors like the signal recognition particle (SRP) and endoplasmic reticulum (ER) translocon components [18]. The occurrence of cytoplasmic foci in plants with elevated RT activity (e.g., *ddm1 Arabidopsis* mutants) has been previously described [19,20]. However, these cytoplasmic components were associated with RT silencing rather than with the promotion of RT replication. Thus, whether retrosomes exist in plant cells remains an open question.

Here we aimed to investigate the structural organization and intracellular localization of the *Arabidopsis thaliana* EVD GAG protein, its aggregation dynamics across diverse plant species, and the transcriptomic consequences of its overexpression. Our findings demonstrate that GAG protein forms retrosome-like cytoplasmic granules in *Arabidopsis* and in distantly related species, *Nicotiana benthamiana* and *Helianthus annuus*. RNAseq analysis of *Arabidopsis* plants overexpressing GAG showed significant up-regulation of stress-responsive genes in wild type and in *ddm1* genetic backgrounds. Altogether, our results revealed that the GAG protein drives the formation of cytoplasmic aggregates and orchestrates broad transcriptomic changes, suggesting a link between LTR retrotransposon activity and stress-related regulatory networks in plants.

## 2. Results

### 2.1. Structure and Intracellular Localization of A. Thaliana GAG Protein

To study plant GAG protein, we used the GAG protein encoded by well characterized EVD LTR retrotransposon (evdGAG) of *Arabidopsis thaliana* [6]. Using AlphaFold3, we visualized evdGAG structure and identified the position of two main structural GAG domains: capsid domain (CA) comprising CA-NTD (N terminal domain) and CA-CTD (C terminal domain) and nucleocapsid (NC) domain (Figure 1a,b) [21]. Previous observations on *S. cereviseae* Ty1 GAG protein revealed an alternative way of capsid formation via prion-like domains found in the region upstream to canonical GAG:GAG interacting CA-NTD and -CTD domains [22]. We did not reveal a prion-like domain in evdGAG, but we found that disorder regions correspond with in-between domain regions of predicted AlphaFold3 evdGAG protein structures (Figure 1c). Thus, most likely evdGAG oligomerization is achieved through CA-CTD interactions of different GAG molecules [23].

To understand the cellular localization of GAG proteins we cloned the intron-less ORF of evdGAG into the expression vector pCambia under the control of the 35S promoter. The N- or C- end of evdGAG was fused with GFP resulting in GFP:evdGAG and evdGAG:GFP, respectively. We used the PEG-mediated transfection of the protoplast method to obtain the transient overexpression of evdGAG protein and to distinguish the GFP fused evdGAG signal location in individual cells. Fluorescent microscopy of the *A. thaliana* protoplasts after GFP fused evdGAG plasmid transfection revealed the presence of multiple large GFP signals (evdGAG:GFP avg. = 7.37 ± 6.1 and GFP:evdGAG avg. = 5.79 ± 4.8) from the evdGAG protein in the cytoplasm (Figure 2a). No visual differences in the cellular localization picture of evdGAG loci were observed for GFP:evdGAG and evdGAG:GFP genetic constructs. We further evaluated the cellular location of the evdGAG in the protoplasts of *ddm1 A. thaliana* mutants possessing natural EVD activity. Similarly to the wild type genetic background, we observed GFP signals in the cytoplasm of the *ddm1* protoplast (evdGAG:GFP avg. = 8.64 ± 4.4 and GFP:evdGAG avg. = 7.61 ± 4.5) (Figure 2b). A comparison of the number of GFP loci in *ddm1* and Col-0 protoplasts revealed no significant differences. We found a significantly higher number of evdGAG loci in *ddm1* protoplasts compared to the protoplast of wild type plants (*p*-value = 4 × 10^6^, Figure 2b, Appendix A). Additionally, we found that larger evdGAG loci tend to be localized in a proximity to the nucleus, resembling ‘retrosomes’, which are structures that were previously described for *Saccharomyces cerevisiae* Ty1 retrotransposon [18,24,25,26].

### 2.2. Intracellular Localization of evdGAG Protein in Distant Plant Species

To achieve a deeper insight into the dependency of the observed evdGAG aggregates on plant species, we performed PEG-mediated 35S:GFP:evdGAG transfection of protoplasts from distant plant species, namely *Nicotiana benthamiana* (*N. benthamiana*) and *Helianthus annuus* (*H. annuus*) (Figure 3). Fluorescent microscopy revealed several large evdGAG aggregates in the cytoplasm of the protoplasts. The same as for *A. thaliana*, several evdGAG protein structures were found around the cell nucleus for both plant species, indicating that *GAG* aggregate formation resulted from physicochemical properties of GAG protein rather than from host factors. We also noted that in *N. benthamiana* and *H. annuus*, protoplast evdGAG signals are often colocalized with the nucleus suggesting that GAG proteins may directly interact with nucleus membrane and/or pass through it.

### 2.3. GAG Overexpression Induces Transcriptional Activation of Stress-Responsive Genes in Wild-Type A.Thaliana and ddm1 Mutant

In order to obtain *GAG* overexpression in *A. thaliana* lines (GAGoe lines), we created genetic constructs (35S:evdGAG) consisting of 35S promoter, evdGAG intron-less ORF tagged with 3xFLAG (Figure 1a). Previously, it was shown that DNA methylation of evdGAG-containing T-DNA in wild type Col-0 plants rapidly occurred within two *A. thaliana* generations [7]. However, it was not clear whether this leads to complete silencing of evdGAG protein expression. To find GAGoe plants, we performed Western blotting analysis of the obtained T_4_ generation of 35S:evdGAG transformants. This analysis identified a plant line with detectable GAG protein after Western blotting of crude protein extracts (Figure 4a) and protein fraction obtained after immunoprecipitation (IP) with anti-flag antibodies. We also performed an *A. thaliana ddm1* mutant floral-dip transformation with the 35S:evdGAG construct. Similarly to GAGoe lines, we could steadily detect the production of GAG protein (Figure 4b) in *ddm1:GAGoe* plants up to T_4_ generation.

To test whether protein overexpression can cause changes in a transcriptome landscape, we performed RNA-seq analysis of WT (Col-0) and GAGoe *A. thaliana* plants. This analysis demonstrated a drastic effect of GAG overexpression on transcriptome composition. Overall, we detected 825 differentially expressed genes (DEGs) including 397 down-regulated genes and 428 up-regulated genes (minimal fold change = 2 and FDR = 0.01) (Appendix A). We next performed enrichment analysis of differentially expressed genes (DEGs) using Gene Ontology (GO) terms and KEGG pathway databases. Among the down-regulated genes, we identified significant enrichment in pathways related to photosynthesis, carbon metabolism, cell cycle regulation, and translation. In contrast, up-regulated genes showed overrepresentation in biological processes associated with plant defense and stress responses, including “response to external biotic stimulus,” “killing of cells of another organism,” and “response to fungus.” Additionally, the up-regulated DEG set was enriched for genes marked by H3K27me3, a repressive histone modification dynamically involved in regulating stress-responsive genes in *Arabidopsis* (PlantGSEAv2 database). GO cellular component analysis revealed that up-regulated DEGs predominantly encode proteins localized in extracellular regions, including the cell periphery, apoplast, cell wall, and plasma membrane. Conversely, down-regulated DEGs were enriched for plastid-localized proteins. Previous studies in yeast demonstrated that retrosomes co-opt P-body components (e.g., Dhh1, Lsm1, Xrn1) to facilitate retrotransposon RNA localization and particle assembly [14]. To investigate potential parallels, we analyzed the expression of Arabidopsis genes encoding core P-body components (DCP1, DCP2, VARICOSE (VCS), DCP5, EXORIBONUCLEASE 4 (XRN4), and TZF1). Notably, none of these genes were significantly up-regulated in GAGoe plants. Instead, AtTZF1 and AtTZF2 exhibited significant downregulation (log2FC = −1.2 to −1.5, adjusted *p* < 0.01). We then examined genes associated with stress granule (SG) formation. Three SG-related genes were up-regulated in the GAGoe plants: Rbp47b (AT3G19130; log2FC = 1.5, adjusted *p* = 3.08 × 10^−5^), AT1G12010 (log2FC = 3.6, adjusted *p* = 2 × 10^−6^), and AT5G64120 (log2FC = 2.2, adjusted *p* = 2 × 10^−6^). These findings suggest that GAG overexpression induces transcriptional activation of stress granule-associated genes and other stress-responsive genes encoding extracellular proteins, while repressing chloroplast-related processes.

We investigated whether similar transcriptome changes observed in GAGoe plants also occur in *ddm1* mutants, which naturally express endogenous GAG proteins. RNA-seq analysis comparing *ddm1* and wild-type plants identified 828 differentially expressed genes (DEGs): 696 up-regulated and 132 down-regulated. GO enrichment analysis revealed significant overrepresentation of cell wall biogenesis-related genes among up-regulated DEGs and response to bacterium and salicylic acid signaling pathways among downregulated DEGs. These findings indicate distinct transcriptomic profiles in *ddm1* compared to WT, likely due to the pleiotropic effects of the *ddm1* mutation.

To assess the impact of GAG overexpression in the ddm1 background, we analyzed *ddm1* GAGoe transformants versus ddm1 plants, identifying 443 DEGs (148 up-regulated, 295 down-regulated)—the lowest number of DEGs across all comparisons. Only 22 DEGs overlapped between WT GAGoe and ddm1 GAGoe plants. Despite this limited overlap, GO enrichment analysis showed collinear trends, with defense response to other organisms being a top up-regulated term in both genotypes. Search of common genes between WT and *ddm1* background, whose expression was altered by GAGoe, revealed several genes connected with biotic stimuli response such as APK4 (sulfation of secondary metabolites, including the glucosinolates), AT3G09450 (fusaric acid resistance family protein), GLUTAREDOXIN 3 (operates downstream of cytokinins in a signal transduction pathway), RLP33 (receptor like protein 33), AT4G10290 (RmlC-like cupins superfamily protein), AT1G22570 (major facilitator superfamily protein).

In conclusion, our results demonstrated that GAG protein generates cytoplasmic aggregates and orchestrates broad transcriptomic changes, directly linking LTR retrotransposon activity to stress related regulatory networks in plants.

These results suggest that GAG overexpression consistently activates biotic stress response pathways, irrespective of the genetic background (ddm1 or WT).

## 3. Discussion

Our study demonstrates that the evdGAG protein forms distinct cytoplasmic aggregates in *Arabidopsis thaliana* protoplast cells. Notably, larger GAG aggregates were frequently localized near the nucleus, suggesting that these structures may represent retrosomes-specialized sites for virus-like particle (VLP) assembly and retrotransposon mobilization. This localization pattern resembles that of *Saccharomyces cerevisiae* Ty1 GAG, which accumulates in the perinuclear region during VLP maturation, potentially through interactions with the endoplasmic reticulum (ER) [18]. The consistent localization of evdGAG aggregates near the nucleus across distantly related plant species such as *Nicotiana benthamiana* and *Helianthus annuus*, implying that retrotransposon-associated pathways might share common regulatory components across diverse plant taxa. Furthermore, the lack of significant variation in evdGAG localization between species indicates that its intracellular trafficking is largely independent of species-specific cellular environments, reinforcing the idea of a universally conserved LTR retrotransposon life cycle within plant cells.

Our RNA-seq analysis revealed that overexpression of evdGAG alters the transcriptional landscape in both wild-type and ddm1 mutant backgrounds. Notably, GAG overexpression leads to upregulation of stress-responsive genes, including those involved in the formation of stress granules (SGs). SGs form in response to cellular stress, sequester mRNAs and RNA-binding proteins, and are typically associated with translational arrest [27]. Importantly, SGs also play a role in sequestering retrotransposon RNA-protein complexes (e.g., LINE-1 RNPs), thereby inhibiting retrotransposon activity by preventing reverse transcription, nuclear import of retrotransposon RNA, and integration of transposable element (TE) cDNA into the genome [19]. This sequestration functions as a host defense mechanism to limit retrotransposition and maintain genomic stability [19,28,29]. The connection between SGs and GAG has not been established for retrotransposons in plants. However, recent studies have shown that GAG proteins can trigger stress granule formation in some retroviruses such as HIV-2, whereas in others like HIV-1 and Human T-lymphotropic virus-1 (HTLV-I), GAG actively blocks or disassembles stress granules [30,31]. The latter represents a viral counter-defense mechanism against SG-mediated inhibition of retrovirus replication. Whether a similar host defense mechanism operates in plant cells remains an open question. It may be suggested that the upregulation of stress-responsive genes observed in our GAG overexpression lines reflects a cellular response to retrotransposon hyperactivity in plants. However, the opposite scenario also exists: interactions between SG proteins and retrotransposon components can facilitate RNA handling and assembly. For example, elevated levels of HIV-2 Gag result in its localization to stress granules, even in the absence of genomic RNA [30]. Moreover, SGs can indirectly promote the HIV-1 life cycle by providing Staufen1 proteins that assemble into Staufen1 ribonucleoprotein complexes (SHRNPs), which facilitate viral RNA packaging and GAG multimerization [32]. Whether the observed stress-related transcriptome responses and SG-like GAG aggregates similarly assist plant LTR retrotransposons in completing their life cycle remains to be determined in future studies.

## 4. Materials and Methods

*EVD_GAG* genes cloning

pBI121 was used as a vector for intron-less *evdGAG* cloning tagged with 3xFLAG epitope from both N- and C-terminus. pCambia was used as a vector for intron-less *evdGAG* cloning tagged with eGFP from both N- and C-terminus. PCR reactions were performed with Phusion^®^ High-Fidelity DNA Polymerase (NEB, Ipswich, MA, USA) according to the instructions provided by the manufacturer. Obtained product was purified with the Evrogen CleanUp Standard Kit (Evrogen LLC, Moscow, Russia). Restriction digestion was carried out with FastDigest™ enzymes (Thermo Fisher Scientific, Waltham, MA, USA) according to manufacturer’s instructions. Restricted DNA product was analyzed by electrophoresis and purified from gel with Evrogen CleanUp Standard Kit (Evrogen LLC, Moscow, Russia) Primers for cloning and screening are stated in the Appendix A. Colony PCR reactions were performed with Encyclo DNA Polymerase (Evrogen LLC, Moscow, Russia) according to the instructions provided by the manufacturer.

*AtEVD_GAG* overexpression plant hybrids obtainment

*A. thaliana 35S:evdGAG* plant lines were obtained via floral-dip transformation of *A. thaliana* WT Col-0 ecotype. Floral-dip transformation was performed via standard Zhang et al. protocol [33]. T_1_ transformants were subjected to full-genome Oxford Nanopore sequencing in order to find the site of t-DNA insertion. Similarly, floral-dip transformation was performed with ddm1 KO mutants to obtain *ddm1:35S:evdGAG* hybrid lines. T_1_ progeny was screened for both homozygous t-DNA insertion and circular EVD-born cDNA presence. Primers used for screening are stated in Appendix A.

PEG-mediated transient gene expression in plant protoplasts

The general procedure was performed according to Mathur and Koncz protocols with minor changes [34]. Plant protoplasts were extracted from 3–4-week-old in vitro grown plants. Leaves were collected, cut with a razor blade, and incubated for 16 h 28 °C in protoplast isolation enzyme solution without rotation in dark conditions. Obtained floating protoplasts were filtered through nylon mesh and centrifuged at 1000 rpm for 5 min. Floating protoplasts were transferred to a new tube and washed with W5 solution twice and then resuspended in 1 mL W5 and the protoplasts were counted using a hemocytometer. Prior to transformation, cells were washed in Mg/Ma solution and cell number was adjusted to 3 × 10^6^ mL^−1^. The 10^6^ protoplasts (in 0.3 mL) were transferred to a 15 mL centrifuge tube. A total of 50 mg plasmid DNA plus 50 mg sheared carrier DNA (e.g., herring sperm or salmon sperm DNA) was added. Then, a total of 0.3 mL PEG solution was added, mixed very gently by rotating the tube, and left for 30 min in dark. This was diluted very slowly with 10 mL of W5 over approx. 20 min, a drop at a time. The protoplasts were pelleted by centrifugation at 600 rpm for 5 min. Then, the protoplasts were resuspended in ~10 mL of W5 and left in the Petri dish for 24–48 h 28 °C with gentle agitation. After 24–48 h incubation, protoplasts were visualized via fluorescent microscopy (Appendix A). Statistical analysis of microscopy data was performed using an R script with the rstatix package. Comparisons were conducted using ANOVA followed by pairwise *t*-tests with Bonferroni correction. Additionally, statistical comparisons were performed using the Wilcoxon test with Bonferroni correction. All statistical data are provided in Appendix A.

Differential gene expression analysis

For RNA sequencing, Arabidopsis Col-0 plants were grown on 1/2 MS medium. RNA was isolated using the ExtractRNA kit (Evrogen, Moscow, Russia) according to the manufacturer’s instructions. Poly-A+ RNA sequencing was performed with assistance of BGI (Shenzhen, China) on DNBSEQ (DNBSEQ Technology) platform. Three biological replicates were analyzed for each sample (Appendix A). The raw reads were pre-processed using BGI-Tech bioinformatics workflow. Reads were mapped to TAIR10 genome using HISAT2 with default settings [35]. The resulting alignment files were processed with SAMtools for sorting and indexing [36]. Gene-level quantification was conducted using featureCounts to obtain raw read counts per gene [37]. Initially, gene expression data were uploaded and pre-processed within iDEP, which included filtering lowly expressed genes by CPM values (min 0.5), normalization, and gene ID conversion to Ensembl identifiers to ensure compatibility with internal annotation databases covering numerous species. Differential gene expression analysis was subsequently performed using DESeq2 implemented within the iDEP platform (Appendix A) [38,39].

Gene Ontology (GO) and KEGG Pathway Enrichment Analysis

Gene Ontology (GO) and Kyoto Encyclopedia of Genes and Genomes (KEGG) pathway enrichment analyses were conducted using the iDEP web application [39]. For functional enrichment, iDEP performed GO enrichment analysis to identify overrepresented biological processes, molecular functions, and cellular components among the DEGs. KEGG pathway analysis was also conducted to determine significantly enriched metabolic and signaling pathways. These analyses leveraged centralized annotation databases from Ensembl and KEGG, with pathway visualization supported by the Pathview package integrated in iDEP.

GAG protein purification

GAG protein purification was conducted in accordance with the Song et al. protocol [40]. A total of 1 g of 2 weeks ddm1 and 35S:EVD_GAG-ddm1 seedlings in two repeats was used for protein extraction and 1 g of powder was lysed in 1 mL of lysis buffer. After double 4 °C 13,000× *g* 10 min centrifugation, equilibrated Biolegend Anti-FLAG (L5) affinity gel was added to the purified crude extract. Incubation of affinity gel in crude extract took 2 h in 4 °C with gentle rotation. Afterwards, the gel was centrifuged at 4 °C 1000× *g* for 2 min and washed with lysis buffer three times. Afterwards, affinity gel was analyzed on SDS-PAGE and Western blot Appendix A.

## Figures and Tables

**Figure 1 plants-14-01894-f001:**
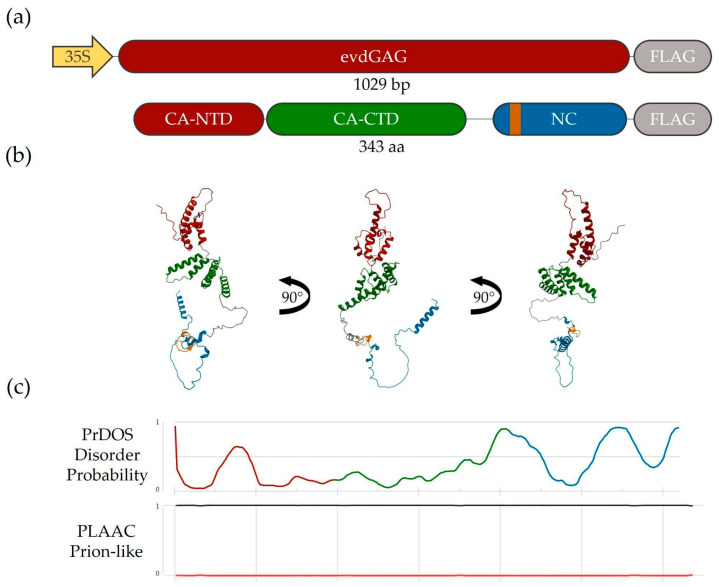
Structural features of EVD GAG protein. (**a**)—Schematic representation of genetic construct used for *A. thaliana* transformation and evdGAG protein domain organization. 35S—35S promoter, evdGAG—evdGAG gene ORF, FLAG—3xFLAG tag, CA-NTD—N-terminal capsid protein domain, CA-CTD—C-terminal capsid protein domain, NC—nucleocapsid protein domain; (**b**)—evdGAG protein structure predicted by AlphaFold3 tool. Red—CA-NTD domain; green—CA-CTD domain; blue—NC domain, orange—Zn-knuckle region. (**c**)—Protein disorder region and prion-like domain prediction results for evdGAG protein made via PrDOS and PLAAC tools, respectively.

**Figure 2 plants-14-01894-f002:**
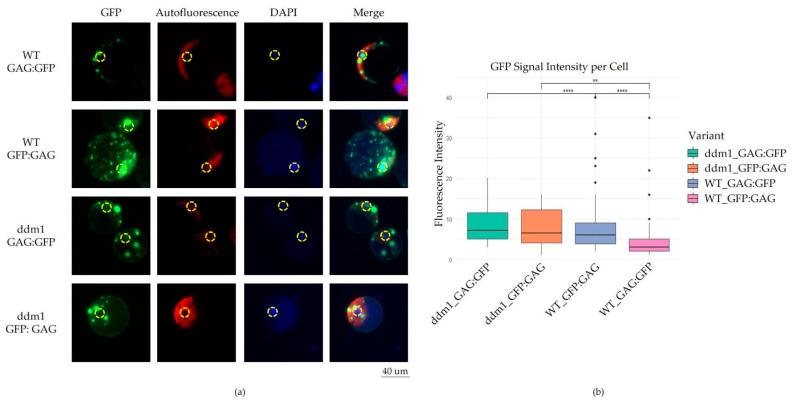
Fluorescent microscopy of evdGAG protein fused with GFP tag: (**a**) Visualization of evdGAG protein structures in *A. thaliana* cell cytoplasm. Vertical columns indicate fluorescence microscopy filters: GFP, chlorophyll autofluorescence, DAPI fluorescence and merged image of all filters. Yellow dashed circles indicate nuclei position in cells. Horizontal rows indicate genetic background of obtained protoplasts (WT or ddm1) and type of vector that was used for PEG-mediated transformation (*GAG:GFP* or *GFP:GAG*). (**b**) Statistical analysis of obtained fluorescent microscopy. X-axis indicates a variant of the protoplast transformation. Y-axis indicates the number of evdGAG protein structures per individual cell. Box plot color indicates type of protoplast transformation experiment: green—ddm1_GAG:GFP; orange—ddm1_GFP:GAG; blue—WT_GAG:GFP; pink—WT_GFP:GAG. Statistical significance between variants is indicated with asterisk symbols (“**”—*p* ≤ 0.01, “****”—*p* ≤ 0.0001).

**Figure 3 plants-14-01894-f003:**
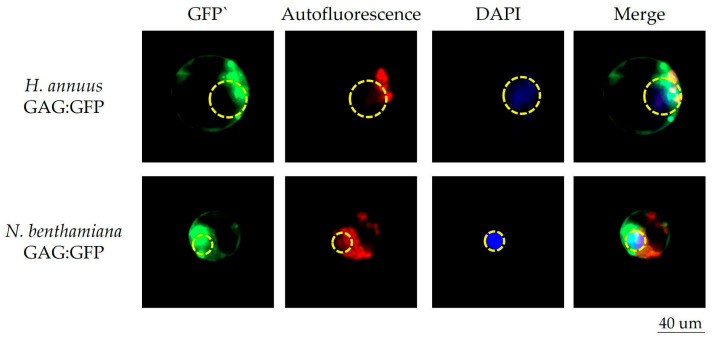
Fluorescent microscopy of evdGAG protein fused with GFP tag: Visualization of evdGAG protein structures in different plant species cell cytoplasm. Vertical columns indicate fluorescence microscopy filters: GFP, chlorophyll autofluorescence, DAPI fluorescence, and merged image of all filters. Yellow dashed circles indicate nuclei position in cells. Horizontal rows indicate genetic background of obtained protoplasts (*H. annuus* or *N. benthamiana*) and type of vector which was used for PEG-mediated transformation (*GAG:GFP* or *GFP:GAG*).

**Figure 4 plants-14-01894-f004:**
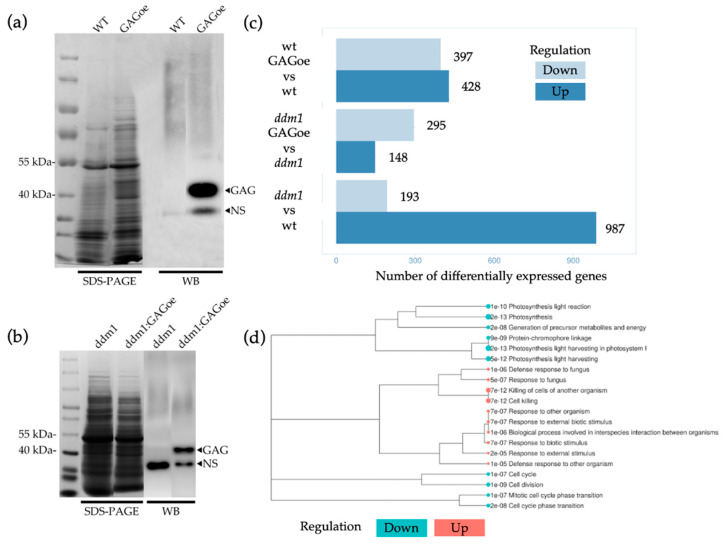
Western blot and RNAseq analysis of GAGoe lines. (**a**,**b**) SDS-PAGE (two left lines) and Western blot (two right lines) results; (**c**)—Comparison of the number of differentially expressed genes identified by RNAseq analysis; (**d**)—GO enrichment analysis of DEGs from RNAseq comparison of GAGoe vs. Col-0 lines.

## Data Availability

The RNAseq data produced for this study are available in the Sequence Read Archive (SRA), NCBI, under Bioproject Accession PRJNA1273727.

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
