# Peer review of "GAG Protein of Arabidopsis thaliana LTR Retrotransposon Forms Retrosome-like Cytoplasmic Granules and Activates Stress Response Genes"

_plants, 2025, doi:10.3390/plants14131894_

Round 1

Reviewer 1 Report

Comments and Suggestions for Authors

The manuscript presents interesting results concerning the biology of retrotransposons. Overall, it is well-written, and the data presented are mostly supported by appropriate results. However, the take-home message of the paper is formulated based on RNA-seq data analysis, yet the Materials and Methods section does not even mention the number of biological replicates used for this analysis. Furthermore, there is no information provided on how the GO and KEGG analyses were conducted.

The RNA-seq data should be deposited in a public database, and the link or ID allowing access to them should be added to the Materials and Methods. If the data are deposited in the SRA, as mentioned in line 353 (Data Availability Statement: "The RNA-seq data produced for this study are available in the Sequence Read Archive (SRA), NCBI, under Bioproject Accession PRJNAXXXX"), then the Bioproject ID must be completed and the SRA IDs should be provided in an additional Supplementary Table summarizing the RNA-seq data (including sequencing depth, number of mapped reads, and identifiers allowing access to the RNA-seq results for each individual sample). Also, It would be good to include a PCA plot based on expression data to show how the samples are grouping.

It is unclear what is presented in Supplementary Data S1. The title states that it contains DEGs, but it seems that the majority of annotated Arabidopsis genes are listed there. Starting from row 233343, all fields are filled with "NA." Are these genes not expressed at all? The column names are also unclear. For example, does a name like "ddm-wt" represent expression change in ddm compared to wt? Do columns 10–21 contain expression values for individual samples? What is the unit—TPM? It should be included in the column name. Were the values calculated as means from biological replicates?If the table contains both expression levels and expression changes, then the title should be changed from “List of differentially expressed genes (DEGs)” to “Expression levels and expression changes among tested samples.” The meaning of “NA” values should be clearly explained.

What statistical test is used in Supplementary Data S3? The measurement results and the results of the statistical analysis should be placed in two separate tables, each with a clear and descriptive title.

There are also some typo errors in the manuscript, e.g., in line 62: “VL)” that should be corrected.

Author Response

Dear Reviewer,

 We greatly appreciate your efforts in improving our manuscript. We replied to all comments individually and made the corresponding changes to the MS. 

Q1: The take-home message of the paper is formulated based on RNA-seq data analysis, yet the Materials and Methods section does not even mention the number of biological replicates used for this analysis.

 A1: Indeed, this information was not included in the MS. Three biological replicates were analysed for each sample. This information has been added to the MS.

Q2: Furthermore, there is no information provided on how the GO and KEGG analyses were conducted.

 A2: GO and KEGG enrichment analysis was performed in iDEP web server. We added new section in M&M with the analysis description.

Q3: The RNA-seq data should be deposited in a public database, and the link or ID allowing access to them should be added to the Materials and Methods. If the data are deposited in the SRA, as mentioned in line 353 (Data Availability Statement: "The RNA-seq data produced for this study are available in the Sequence Read Archive (SRA), NCBI, under Bioproject Accession PRJNAXXXX"), then the Bioproject ID must be completed and the SRA IDs should be provided in an additional Supplementary Table summarizing the RNA-seq data (including sequencing depth, number of mapped reads, and identifiers allowing access to the RNA-seq results for each individual sample). 

 A3: The RNAseq reads were deposited to the NCBI and the following information has been added to the MS: ‘The RNAseq data produced for this study are available in the Sequence Read Archive (SRA), NCBI, under Bioproject Accession PRJNA1273727.’ We also added a new supplementary table with the required information: ‘Supplementary table S6. FASTQ RNAseq reads accessions for sequencing samples                                                    ’

Q4: It is unclear what is presented in Supplementary Data S1. The title states that it contains DEGs, but it seems that the majority of annotated Arabidopsis genes are listed there. Starting from row 233343, all fields are filled with "NA." Are these genes not expressed at all? The column names are also unclear. For example, does a name like "ddm-wt" represent expression change in ddm compared to wt? Do columns 10–21 contain expression values for individual samples? What is the unit—TPM? It should be included in the column name. Were the values calculated as means from biological replicates?

 A4: The table has been renamed to “Supplementary Table S1. Differential expression results for:

  • ddm1 vs wild type (ddm1-WT)
  • ddm1 overexpressing GAG vs ddm1 (ddm1OE-ddm1)
  • Col-0 overexpressing GAG vs wild type Col-0 (gagOE-WT).

“NA” (Not Available) denotes cases where DESeq2 could not generate statistically robust results due to insufficient expression variance or low read counts. The final columns provide normalized read counts (counts per million, CPM) calculated during iDEP’s pre-processing step. This clarification has been added to the table legend and Methods section.

Q5: If the table contains both expression levels and expression changes, then the title should be changed from “List of differentially expressed genes (DEGs)” to “Expression levels and expression changes among tested samples.” The meaning of “NA” values should be clearly explained.

A5: The corrections have been made.

Q6: What statistical test is used in Supplementary Data S3? The measurement results and the results of the statistical analysis should be placed in two separate tables, each with a clear and descriptive title.

A6: Statistical analysis was performed via R script using the rstatix package. Statistical comparison was done by ANOVA test followed by pairwise t-test with bonferroni correction. Statistical comparison was followed by the Wilcoxon test (bonferroni correction). New Supplementary Data S5 was added and all statistics were moved there. Supplementary Data S3 was renamed.

Q7: There are also some typo errors in the manuscript, e.g., in line 62: “VL)” that should be corrected.

 A7: Thank you for pointing out the typographical errors in the manuscript. We have carefully reviewed the text and corrected the error on line 62 (“VL)”) as well as other minor typographical mistakes throughout the manuscript.

Reviewer 2 Report

Comments and Suggestions for Authors

The authors investigate the structural properties, subcellular localization, and transcriptomic effects of the Arabidopsis LTR retrotransposon EVD encoded GAG protein. Using AlphaFold3 structural modeling, fluorescent microscopy in protoplasts, and RNA-seq analysis, the authors showed a lot of new characterizations of evdGAG.

Major:

  1. The evidence might be not sufficient enough to link LTR activity to the differential expression including the up-regulation of stress related genes it causes: It may be just because the translated GAG protein, especially if expressed at high levels and forming cytoplasmic aggregates, could act as a foreign or misfolded protein, which the host cell interprets as a form of proteotoxic stress. This in turn could trigger cellular defense responses, including activation of stress granule-associated genes, upregulation of extracellular defense genes and downregulation of translation and photosynthesis. So, it remains unclear whether this reflects retrotransposon mobilization or merely a protein-level effect.

  1. TEs are highly repetitive sequences in the Arabidopsis genome. The authors used short-read sequencing to analyze their dynamics. But I didn’t see any information in the Methods section that introduces how to process multiple aligned reads during the quantification step. Please add more details in the bioinformatics section, including parameter settings.
  2. It is better to consider long-read sequencing (ONT/PacBio) with TE capture methods to assess the full-length LTR retrotransposon structure and activity directly.

Minor:

  1. Fluorescence images in Figures 2 and 3 would be better if they clearly indicate nucleus boundaries in merged images.
  2. There are minor repetitions, such as “stress stress-responsive genes” (line 154).

Author Response

Dear Reviewer,

 We greatly appreciate your efforts in improving our manuscript. We replied to all comments individually and made the corresponding changes to the MS. 

Major:

Q1: The evidence might be not sufficient enough to link LTR activity to the differential expression including the up-regulation of stress related genes it causes: It may be just because the translated GAG protein, especially if expressed at high levels and forming cytoplasmic aggregates, could act as a foreign or misfolded protein, which the host cell interprets as a form of proteotoxic stress. This in turn could trigger cellular defense responses, including activation of stress granule-associated genes, upregulation of extracellular defense genes and downregulation of translation and photosynthesis. So, it remains unclear whether this reflects retrotransposon mobilization or merely a protein-level effect.

A1: We agree that the observed transcriptomic changes in response to GAG overexpression in ddm1 and Col-0 could reflect a general plant cell response to protein aggregate formation and proteotoxicity. While this transcriptome changes could also occur during LTR retrotransposon hyperactivity (as discussed in our HIV-based examples), we agree that establishing a direct causal link between LTR retrotransposon activity and the observed broad transcriptomic changes requires further investigation. We have revised our conclusions to reflect this nuance and emphasize the need for future studies to disentangle these mechanisms.

 Q2: TEs are highly repetitive sequences in the Arabidopsis genome. The authors used short-read sequencing to analyze their dynamics. But I didn’t see any information in the Methods section that introduces how to process multiple aligned reads during the quantification step. Please add more details in the bioinformatics section, including parameter settings.

A2: In this study, we did not perform expression analysis of TEs and only estimated differential expression for annotated genes. Therefore, we used the default settings for HISAT2. We have added this information to the Materials and Methods section.

Q3: It is better to consider long-read sequencing (ONT/PacBio) with TE capture methods to assess the full-length LTR retrotransposon structure and activity directly.

A3: We agree that long-read sequencing is much better suited for obtaining information on full-length LTR retrotransposon transcripts. However, this was not the aim of our study within the scope of this manuscript.

Q4: Fluorescence images in Figures 2 and 3 would be better if they clearly indicate nucleus boundaries in merged images.

A4: Thank you for this comment. We corrected the Figures. 

Q5: There are minor repetitions, such as “stress stress-responsive genes” (line 154).

 A5: Thank you for pointing out the typographical errors in the manuscript. We have carefully reviewed the text and corrected the error as well as other minor typographical mistakes throughout the manuscript. 

Round 2

Reviewer 1 Report

Comments and Suggestions for Authors

-

Reviewer 2 Report

Comments and Suggestions for Authors

all the questions have been answered